# The Influence of Early Nutrition on Brain Growth and Neurodevelopment in Extremely Preterm Babies: A Narrative Review

**DOI:** 10.3390/nu11092029

**Published:** 2019-08-30

**Authors:** Barbara E. Cormack, Jane E. Harding, Steven P. Miller, Frank H. Bloomfield

**Affiliations:** 1Starship Child Health, Auckland City Hospital, Auckland 1023, New Zealand; 2Liggins Institute, University of Auckland, Auckland 1142, New Zealand; 3Department of Paediatrics, Hospital for Sick Children, Toronto, ON M5G, Canada; 4Department of Paediatrics, University of Toronto, Toronto, ON M5S, Canada

**Keywords:** nutrition, parenteral nutrition, amino acid, extremely low birthweight, preterm, newborn, gestational age, growth, infant, premature, nutrition disorders/etiology, follow up, brain

## Abstract

Extremely preterm babies are at increased risk of less than optimal neurodevelopment compared with their term-born counterparts. Optimising nutrition is a promising avenue to mitigate the adverse neurodevelopmental consequences of preterm birth. In this narrative review, we summarize current knowledge on how nutrition, and in particular, protein intake, affects neurodevelopment in extremely preterm babies. Observational studies consistently report that higher intravenous and enteral protein intakes are associated with improved growth and possibly neurodevelopment, but differences in methodologies and combinations of intravenous and enteral nutrition strategies make it difficult to determine the effects of each intervention. Unfortunately, there are few randomized controlled trials of nutrition in this population conducted to determine neurodevelopmental outcomes. Substantial variation in reporting of trials, both of nutritional intakes and of outcomes, limits conclusions from meta-analyses. Future studies to determine the effects of nutritional intakes in extremely preterm babies need to be adequately powered to assess neurodevelopmental outcomes separately in boys and girls, and designed to address the many potential confounders which may have clouded research findings to date. The development of minimal reporting sets and core outcome sets for nutrition research will aid future meta-analyses.

## 1. Introduction

Improving neurodevelopmental outcomes after extremely low birth weight (ELBW, birth <1000 g) and extremely preterm birth (<28 weeks’ gestational age) is a major challenge in modern neonatal care. Extremely preterm and ELBW babies are born during the late second and third trimesters of pregnancy at a time of critical brain development. In normally developing fetuses, in vivo measurements of total and regional brain volume using advanced magnetic resonance imaging (MRI) techniques showed that from 25 to 37 weeks’ gestation, total brain and cerebral volumes increased by 230%, brain stem volume increased by 134%, and the volume of the cerebellum increased almost four-fold (384%) [1]. From 24 weeks’ gestation, cortical grey matter matured, radial glia disappeared, the complexity of connections increased and cortical folding and gyrification became increasingly complex [2]. During this phase of brain maturation, axons, glial cells, oligodendrocytes, and neurons are also developing rapidly in the white matter [3]. Preterm babies, especially those born ELBW, have high rates of brain injury, which reflects the vulnerability of these developing cell types [4]. Preterm neonates also have evidence of impaired brain maturation, as reflected in altered brain size, structure [5], connectivity and function [6,7] when compared with term-born babies. 

Suboptimal neurodevelopmental outcomes continue to affect 20–45% of preterm babies and up to 50% of those born ELBW [8]. Extremely preterm and ELBW children have deficits relative to their term-born peers in IQ, expressive and receptive language skills, spatial reasoning, motor and visual-motor skills, and executive functioning [9,10,11,12,13,14,15,16,17]. Wide variation occurs in functional impairment, even in children born at similar gestational ages [18].

For extremely preterm babies, reported rates of cerebral palsy vary from as high as 17% in the United Kingdom [19] and 12% in North America [20] to around 10% in Norway [21] and Sweden [22], and as low as 6% to 7% in Japan [23], New Zealand and Australia [24]. A meta-analysis of 30 studies of extremely preterm babies born since 2006 reported that the pooled prevalence of cerebral palsy was 10.0% (95% CI 8.1–12.2) [25], which is substantially lower than the 14% prevalence reported in a similar meta-analysis 10 years earlier [26]. However, the overall prevalence of cognitive impairment and other motor impairments has not decreased (Figure 1).

The prevalence of cerebral palsy rises with decreasing gestational age [25] and changes in the severity of cerebral palsy have also been observed. In a US neonatal network, Adams-Chapman et al. reported that severe cerebral palsy had decreased by 43% from 2011 to 2015, whereas mild cerebral palsy increased by 13%. However, there was no change over time in the incidence of neurodevelopmental impairment [20]. 

Pascal et al. reported that the prevalence of motor delays was 45% among children born extremely preterm and 34% for those born ELBW [25]. For extremely preterm babies, the reported rate of blindness was 0.4–2% and hearing impairment was 1–3% [20,21,22,24]. 

Children born ELBW have lower cognitive scores at school entry age in comparison with term-born counterparts [27]. Up to 50% have significant cognitive, sensory or behavioural deficits [4]. Cognitive impairment is significantly correlated with lower birthweight and gestational age at birth [27,28]. While a child may have disability in several domains, cognitive impairment can occur without significant motor impairment [29]. Furthermore, children with broadly normal IQ scores may have deficits in attention or executive function (cognitive flexibility, inhibitory control, working memory) [30,31] and visual or language processing [32,33] which are still present into adolescence [34,35,36,37]. The rate of behaviour problems in extremely preterm and ELBW children is up to four times higher than in their term born peers and includes attention deficit hyperactivity disorder, internalising symptoms, such as anxiety and depression [28,38,39,40], social functioning difficulties and peer victimisation [41,42] which may affect the transition to adult life in comparison with their peers [34,35,36,37].

Neurodevelopmental outcome is hugely important for babies, parents and caregivers but is also a significant public health consequence due to the number of ELBW babies who now survive with neurodevelopmental disability. While survival without severe or moderate neuromotor or sensory impairment at 2 years has improved between 1997 and 2011, the risk of developmental delay remains high for children born ELBW or extremely preterm [43]. Hence, there is an urgent need to identify how neurodevelopmental disability occurs and find opportunities to promote optimal brain development and outcomes. In this narrative review, we aimed to summarize current knowledge on how nutrition, and in particular, protein intake, affects neurodevelopment in extremely preterm babies and to suggest strategies to improve future nutrition research in this area.

## 2. Effects of Preterm Birth on the Brain

Brain injury following extremely preterm birth is an important cause of neurodevelopmental impairment and is typically reflected in white matter injury (WMI) and dysmaturation, and intraventricular haemorrhage (IVH) [3]. White matter injury is the most common type of brain injury [3], with cystic periventricular leukomalacia (PVL) as its most severe manifestation. While the incidence of PVL has decreased with modern neonatal care, rates of periventricular haemorrhagic infarction have not declined [44,45]. Cystic PVL and periventricular haemorrhagic infarction are both associated with cerebral palsy, epilepsy, major cognitive disability and visual impairments [46], although cystic periventricular white matter damage seen on magnetic resonance imaging (MRI) does not preclude normal cognitive development [47]. The most prevalent pattern of WMI in contemporary cohorts of preterm neonates is diffuse WMI. The hallmark neuropathological feature of diffuse WMI is dysmaturation of the oligodendroglial lineage, mediated through a specific bioactive hyaluronan fragment that chronically blocks oligodendrocyte progenitor cell maturation and myelination [48]. Diffuse WMI is also associated with disruptions of thalamo-cortical connectivity and neuronal dysmaturation in the basal ganglia and cerebral cortex. Thus, diffuse WMI and associated neuronal dysmaturation is thought to be responsible for the range of mild to moderate cognitive, attentional, behavioural, and social emotional problems that many preterm-born children display [3,49]. There is increasing evidence linking white matter injury and brain maturation as measured by MRI with subsequent neurodevelopmental outcome [50,51,52].

Intraventricular haemorrhage (IVH) typically occurs in the first days after birth. Severe IVH may be associated with periventricular haemorrhagic infarction. Another form of brain injury is cerebellar haemorrhage, which may be isolated or coincident with IVH and PVL [53]. Reduced cerebellar volume at term-equivalent age in preterm babies is seen in association with haemorrhagic parenchymal infarction, IVH with dilatation, and PVL [54]. Cerebellar injury and IVH are more likely to cause milder motor deficits, cognitive and behavioural problems [55,56]. 

## 3. Postnatal Factors that Play a Role in Neurodevelopment 

Multiple factors such as gestational age at birth, size for gestational age, brain injury, growth, nutrition, neonatal morbidities [57,58,59], parental education [60,61], and other social determinants of health influence neurodevelopmental outcomes [62], but only a few of these factors are modifiable during neonatal care. Both sepsis and necrotising enterocolitis (NEC) put preterm babies at greater risk of motor impairment at 2 years [20,63,64], likely mediated by white matter injury [65]. Important causes of white matter injury are inflammation and ischaemia [3,66]. Both are common in preterm babies and initiate the three leading events that cause white matter injury: activation of microglia, free radical attack, and excitotoxicity [66]. Bacterial sepsis activates toll-like receptors that are present on the surface of microglia in white matter [66]. The activation of microglia leads to a release of free radicals as well as pro-inflammatory cytokines, and injury to the developing pre-oligodendrocytes, axons, and neurons in the white matter [66]. The activation of pro-inflammatory cytokines suppresses the synthesis of growth factors, such as insulin-like growth factor (IGF)-1, which are important for brain growth and differentiation [67]. Infection and systemic inflammation also are often associated with haemodynamic instability, resulting in reduced cerebral blood flow due to impaired cerebrovascular autoregulation.

Chronic lung disease also plays a role. A meta-analysis of 71 studies (7752 extremely or very preterm and 5155 full-term control children born between 1990 and 2008) found that chronic lung disease explained 65% of the variance in intelligence across studies [68]. This may be due to cells responding to environmental stress, such as hypoxia or low energy, by down-regulating energy-demanding processes such as protein synthesis and arresting brain development [69], or due to the chronic inflammation associated with chronic lung disease. Alternatively, it may be that there are specific factors that predispose to chronic lung disease that also are detrimental to neurodevelopment. Therefore, strategies to reduce chronic lung disease, systemic infections and inflammatory responses potentially would help prevent white matter injury and promote brain development.

## 4. Early Nutrition Affects Morbidity and Neurodevelopment 

One potential strategy for reducing neonatal morbidity is improved early nutrition. Postnatal undernutrition, specifically inadequate energy in the first 4 weeks after birth, is an independent predictor of chronic lung disease [70]. Early enteral feeding has the potential to influence clinical outcomes, such as NEC and late-onset infection, by influencing the functional adaptation of the gastrointestinal tract and disrupting patterns of adverse microbial colonisation [71]. Delayed enteral feeding and intolerance due to intestinal dysmotility increase the time to full enteral feeds and prolong the duration of intravenous nutrition, increasing the risk of infection and metabolic complications that adversely affect growth and development [72]. Enhanced energy and macronutrient intake in the first weeks after birth have also been associated with improved neurodevelopmental outcomes, including improved language scores in very low birthweight (VLBW) babies [73], increased developmental quotient in those born at <28 weeks’ gestation [74] and decreased incidence of brain lesions on MRI [75] in babies born at <30 weeks’ corrected gestational age (CGA). Emerging evidence confirms that optimising nutrition in the first few days after birth is both important [76] and safe [77] and significant efforts have been made to promote optimal long-term development by improving neonatal nutrition and reducing suboptimal growth. Despite these efforts, early recommended nutrient provision to preterm babies is not routinely achieved [78]. Postnatal undernutrition and faltering growth are common and associated with adverse cortical development in the neonatal period [79] and long-term neurodevelopmental outcomes [80].

## 5. The Role of Specific Nutrients in Brain Development

Various nutrients have been specifically associated with brain structure and neurodevelopment (Figure 2). 

Deficits of these nutrients can affect the developing brain and especially early organizational events such as neurogenesis, cell migration and differentiation [81] and glial cell function [82,83]. Glucose is the primary energy source for the brain and nervous system. The brain consumes about 20% of total body glucose and brain physiology depends on tight regulation of glucose transport and its metabolism [84]. If glucose is limited, the brain is capable of oxidising lactate and ketone bodies as alternative cerebral fuels. A growing body of evidence shows that lactate sustains neuronal activity during glucose deprivation [85,86,87] and that lactate, rather than glucose, is the preferred energy metabolite for the maintenance of several neuronal functions [84]. Deficits of protein, fat, carbohydrate, energy, iron, copper, zinc, iodine, thiamine, folate, selenium, choline, vitamins A, B12, D, C and long chain polyunsaturated fatty acids directly affect brain anatomy and developmental processes [88]. Of particular note is that the deficiency of each one of these nutrients results in similar developmental sequelae (impaired cognitive, motor, social-emotional, and neurophysiologic development), suggesting that the common underlying feature may be the timing of the nutritional insult, or perhaps, that all of these nutrients are required in sufficient amounts to facilitate optimal growth and brain development.

## 6. Early Growth Is Related to Neurodevelopment 

Achieving growth that is equivalent to that of a fetus of the same post-menstrual age has been the aim of neonatal nutritional care for over four decades [89]. In comparison with this standard, many have reported that faltering growth is universal in preterm babies [90,91]. The term ‘faltering growth’ (previously called ‘failure to thrive’) was historically used in paediatric assessment to describe a slower than expected rate of growth for age and sex, but has now been more precisely defined as a fall of 1 to 3 weight centile spaces (the space between adjacent centile lines), depending on weight centile position, or weight below the second centile for age on UK-WHO growth charts [92]. Faltering growth, though common, is not universal in preterm babies. In the past few years, improvements to neonatal care and attention to intravenous and enteral nutrition have made it possible in some neonatal intensive care units for ELBW babies to achieve growth equivalent to intrauterine rates for weight and head circumference, if not always for length [93,94,95]. Nevertheless, worldwide, many babies continue to have less than optimal postnatal growth. 

Faster weight gain from birth to term-equivalent age is associated with better neurodevelopmental outcomes in children born preterm. This was demonstrated in a classic study by Ehrenkranz et al. in which 495 ELBW babies from a multi-centre cohort study were divided into quartiles of in-hospital growth velocity rates [96]. As the rate of weight gain velocity increased between quartile 1 and quartile 4, from 12.0 to 21.0 g·kg^−1^·day^−1^, the incidence of cerebral palsy, Bayley II Mental and Psychomotor Developmental Indices (MDI and PDI), abnormal neurological examination and neurodevelopmental impairment each decreased by at least 50%, and the need for rehospitalization fell from 63% to 45%. Similar findings were observed as the rate of head circumference growth increased. Others have found greater weight and body mass index (BMI) gain to term-equivalent age also are associated with better neurodevelopmental outcomes. In a group of 613 babies born <33 weeks’ gestational age, assessed at 18 months’ corrected age, Belfort et al. found that every one z-score improvement in weight gain and BMI between 1 week of age and term-corrected age was associated with an increase in MDI of 2.4 and 1.7 points and PDI of 2.7 and 2.5 points respectively [97]. Although these increases are modest, other observational studies have found similar positive associations between early weight gain and neurocognitive outcomes at 18−24 months [98,99], in childhood [100], adolescence [101] and adulthood [102]. 

There is also a positive relationship between length and neurodevelopmental outcome [97,103,104,105]. This may be because lean mass deposition relies on adequate intakes of both protein and energy and linear growth reflects protein accretion and the structural growth of the brain [106]. Faster head circumference, length and body mass index gain from birth to term-equivalent age in VLBW babies is associated with better cognitive and motor outcomes [97], general neurocognitive abilities, executive functioning, and visual memory in young adulthood [102]. Therefore, instead of using the “paediatric” definition of faltering growth described earlier [92], it may be more appropriate to use the recently developed criteria for the diagnosis of malnutrition in preterm babies and neonates from the Academy of Nutrition and Dietetics and the American Society for Parenteral and Enteral Nutrition [107]. This defines mild, moderate and severe malnutrition according to decreases in weight or length z-score (mild: <0.8–1.2; moderate, 1.2–2.0, severe ≥ 2.0), and is based on evidence of the z-score change at which detrimental neurodevelopmental effects are seen. Use of the term “malnutrition” increases awareness that suboptimal growth has long-term consequences and that it is not universal, unavoidable and expected in neonatal care. It is also a reminder that undernutrition is the primary cause of faltering growth and malnutrition in preterm babies and therefore modifiable. 

In VLBW babies assessed at 25 years of age, Sammallahti et al. found that, after adjustment for demographic factors and neonatal complications, only faster head circumference growth between birth and term-corrected age, but not between term-corrected age and 12 months, was significantly associated with improvements in IQ and its components, performance and verbal IQ, with effect sizes ranging between 0.33 and 0.42 standard deviations per z-score increase in head growth [108]. There were no statistically significant associations with faster weight or length growth. Head circumference is a proxy for brain growth, and MRI confirms a strong correlation (*r* = 0.68) between head size and brain tissue volume [109]. Several studies have shown associations between reduced brain volumes, particularly cerebellar volumes, and impaired neurodevelopment at 2 years [110,111,112]. Head circumference at birth is only predictive of poor neurodevelopmental outcome in preterm babies if poor growth persists postnatally [113]. In ELBW babies, achievement of head circumference z-score at or above −2 by term-equivalent age is associated with better cognitive outcome at 5.5 years of age when compared with those with a head circumference z-score below −2 [100]. In fact, for preterm babies, head circumference growth is more closely related to neurocognitive abilities than weight gain [102,114,115]. Avoiding faltering postnatal head growth may reduce neurodevelopmental impairment by allowing cortical development to proceed optimally [79]. In preterm babies, increased energy and protein intake in the first month after birth is associated with improved head growth [94,116,117,118] and cognitive outcome in adolescence [57,101], but studies to assess the effect of enhanced nutrition on brain volumes and neurodevelopmental outcome have yielded inconsistent results [112,119,120]. 

## 7. Evidence of Nutritional Effects on the Brain from Brain Imaging 

The use of brain imaging techniques to examine brain structure and function has great potential to give timely, accurate and objective information about the relationships between nutritional interventions, growth and neurodevelopment. Decreased brain volume at term-equivalent age is related to both white matter abnormalities [121,122] and decreased deep nuclear grey matter volume [79,109]. Preterm babies have high rates of white matter abnormalities which are visible on conventional MRI in the form of loss of volume with enlarged ventricles, delayed cortical maturation and diffusion abnormalities [123]. Diffusion-tensor MRI uses the diffusion of extracellular water molecules to infer the brain’s microstructure [124]. Altered regional diffusion measures in very preterm infants imaged at term-equivalent age have been associated with impaired motor and cognitive development at age seven [125]. Fractional anisotropy (FA) is a measure used in diffusion imaging which reflects fibre density, axonal diameter and myelination in white matter. In the cerebral cortex, the radial organization that is expected early in the third trimester is reflected in high FA that decreases to term-equivalent age. Higher FA in the cortex reflects delayed cortical maturation [79] and a slower increase in white matter FA from birth at 24–32 weeks’ gestation to term-equivalent age has been related to impaired developmental outcomes at 18 months of age [51]. Vinall et al. used diffusion tensor MRI FA to assess babies born at <32 weeks at about 32 weeks’ postmenstrual age and again at term-equivalent age. Slower postnatal growth (defined as weight <10th centile at the second scan) was associated with decreased microstructural development of the cerebral cortex, but not white matter, even after accounting for prenatal growth, neonatal illness, and brain injury [79]. 

Specific nutrients also have differential associations with brain regional growth and function (Figure 3). 

MRI scans of regional brain tissue volumes near birth and near term-equivalent age in babies born at <32 weeks’ gestation showed that a 1% higher red blood cell DHA concentration in the first few weeks after birth was associated with 4.3-fold decreased odds of IVH, improved brain microstructural development, and higher language (7.0 points, 95% CI 1.3–12.7, *p* = 0.017) and motor scores (7.5 points, 95% CI 3.1–11.9, *p* = 0.001) on Bayley III assessment at 30–36 months’ corrected age [126]. Higher near-birth DHA concentration was also associated with larger cortical grey matter, deep grey matter, and brainstem volumes, and higher near-term DHA concentration with larger deep grey matter, cerebellar, and brainstem volumes at term-equivalent age. Larger cortical and deep grey matter, cerebellar, and brainstem volumes at term-equivalent age were in turn associated with improved language scores, and larger cerebellar and brainstem volumes with improved motor scores at 30–36 months’ corrected age [127]. DHA concentrations are altered by the type and amount of lipid in the maternal diet and postnatal intravenous and enteral nutrition [128]. Decreased regional white matter diffusivity at term-equivalent age, suggesting improved maturation of cerebral connective tracts, has also been reported in a randomized trial of VLBW babies who received enhanced intravenous and enteral nutrition [129]. The enhanced nutrition was a combination of higher energy, protein, lipid, DHA and vitamin A. Therefore, the influence of individual nutrients is not yet fully defined. 

Total energy and lipid intakes, especially soon after birth, have been identified as important for optimal brain development. Brain abnormalities in white matter, cortical and deep grey matter and cerebellum and brain growth were scored using the Kidokoro scoring system in babies born at ≤30 weeks’ gestation who underwent MRI at term-equivalent age [75]. Higher energy and lipid intakes in the first 2 weeks after birth were associated with improved scores on MRI. Adjusted models estimated a 10 kcal·kg^–1^·day^–1^ increase in energy or a 0.7 g·kg^–1^·day^–1^ increase in total lipid intake reduced the risk of a severely abnormal MRI at term equivalent age by >60% and was related to enteral rather than intravenous lipid. In another study, higher energy and lipid intakes in the first 2 weeks after birth were associated with more robust growth of subcortical structures, cerebellum and total brain, and accelerated white matter microstructural maturation at term-equivalent age in 49 babies born at <30 weeks’ gestation and imaged serially throughout the neonatal period [130]. Higher energy and lipid intake appeared to mitigate the negative impact of respiratory morbidity on brain development. Furthermore, in this study, psychomotor outcome at 18 months’ corrected age was predicted by brain growth, suggesting that the long-lasting effect of early nutrition on neurodevelopment might be mediated by enhanced early brain growth. Lipid was the macronutrient predominantly associated with MRI metrics and, again, mostly due to enteral intake. After correction for birthweight z-score, gestation at birth, postmenstrual age, white matter injury, and clinical outcome (chronic lung disease, PDA, blood culture-proven sepsis, NEC, IVH, and cerebellar injury), cumulative lipid and energy intakes to 2 weeks after birth were positively related to larger cerebellar volumes. In this study, protein and carbohydrate intakes were also significantly associated with brain growth, although others have reported that brain volumes at term were not related to macronutrient intakes [119,131]. 

A positive association was also found between protein intake, early brain growth and neurodevelopmental outcomes in babies born at <31 weeks’ gestation [132]. Enteral protein, fat, and energy intakes were positively associated with cerebellar volume, basal ganglia and thalami volumes assessed at term-equivalent age. Enteral protein intake was related to larger total brain volume, but longer duration of intravenous nutrition was associated with smaller total brain volume, cerebellar, basal ganglia and thalami volumes, and cortical grey matter volume. Weight gain velocity (g·kg^–1^·day^–1^) during the first 4 weeks also showed a positive association with total brain volume, cerebellar, basal ganglia and thalami volumes, and cortical grey matter volumes. Enteral protein, fat, and energy intakes over the first 28 days were positively associated with FA in the posterior limb of the internal capsule, whereas days of intravenous nutrition showed a negative association with FA. Cumulative total protein intake was positively associated with higher cognitive and motor scores at 2 years’ corrected age but with a relatively small effect size (0.5 cognitive and motor scores per 1 g·kg^–1^·day^–1^ total protein). These findings persisted after controlling for birthweight z-score, gestation at birth, gender, white matter injury, and clinical risk factors, and were independent of illness severity. Interestingly, in this study, the amount of breastmilk, weight gain, head circumference, and length growth in the first 4 weeks were not related to neurodevelopmental outcome at 2 years’ corrected age. Of note is that the reported mean energy and protein intakes (110 ± 14 kcal·kg^−1^·day^−1^ and 2.6 ± 0.2 g·kg^–1^·day^–1^ total protein) and growth (14 ± 5 g·kg^–1^·day^–1^) for weeks 2, 3 and 4 in this study were low in comparison with current recommendations [133,134]. 

Neuronal processing speed, which reflects processes such as synaptic efficacy and myelination, is an important component of brain function and can be assessed by measuring visual evoked potentials—electroencephalographic recordings of the brain’s response to a specific visual stimulus [135]. Higher lean mass has been associated with faster neuronal processing, as measured by visual evoked potentials, and may be due to the positive effects of protein status on neuronal growth and differentiation [136]. This is not surprising, because lean mass is an indicator of protein accretion and organ growth, including the brain.

After a randomized trial of 424 preterm babies <1850 g at birth given preterm formula containing higher amounts of protein, carbohydrate, fat, and micronutrients designed to meet their calculated nutritional needs in the first month after birth, follow-up scans and IQ data were obtained from 76 participants in adolescence. Boys who had the higher nutrient formula were found to have significantly greater caudate nucleus volumes at 16 years of age, compared with boys given standard term formula [120]. However, in another trial of 109 smaller babies <29 weeks’ gestation randomized to either an enhanced intravenous and enteral nutrition, or a standard feeding regimen from birth to 34 weeks’ postmenstrual age, quantitative MRI findings (*n* = 65), MDI and PDI were not different between the two groups at 9 months’ corrected age [137]. The actual nutritional intakes of the intervention group in this trial were not very different from the control group, perhaps explaining the lack of effect. Interestingly, total brain volume at 40 weeks’ post-menstrual age, and MDI and PDI at 3 months post-term (*n* = 81) were negatively correlated with cumulative energy deficit (defined as <120 kcal·kg^−1^·day^−1^) at 4 weeks of age, suggesting that preventing early energy deficit promotes neurodevelopment. In this study, 80% of the participants were still in a cumulative protein/energy deficit at the end of 4 weeks. Twenty (24%) of those in deficit at 4 weeks had a head circumference >2 z-scores below the mean at 36 weeks’ post-menstrual age, but none of the babies who were not in deficit. 

Despite many observational studies consistently reporting positive associations between postnatal weight or head growth and neurodevelopmental outcomes, the few intervention trials that aimed to promote postnatal growth in preterm babies have produced little evidence of any beneficial effects of faster early growth on later neurodevelopment [138]. 

## 8. Protein and Amino Acid Requirements

Protein is the major structural and functional component of all cells in the body and consists of chains of amino acids joined together by peptide bonds [139]. The accretion of body protein is dependent on an optimal intake of essential and non-essential amino acids and sufficient energy [140]. Although recommended nutritional intakes are expressed as protein, the biological requirement is for amino acids. Essential amino acids for humans are isoleucine, leucine, valine, lysine, methionine, phenylalanine, threonine, tryptophan, and histidine. Due to immature metabolic processes, several other amino acids are conditionally essential for preterm babies: arginine, glutamine, glycine, proline, taurine, and tyrosine [141,142]. 

Amino acids are also required to synthesize other important nitrogen-containing compounds, such as creatine and peptide hormones [143], and are key components of crucial signalling molecules such as neurotransmitters [81]. For instance, the protein kinase mTOR (mechanistic target of rapamycin) is a crucial cellular signalling hub that integrates internal and external cues, including insulin, growth factors such as IGF-1 and IGF-2 [144], and amino acids, and senses cellular nutrient, oxygen, and energy levels [145] to control cell growth, motility, proliferation and survival, protein synthesis, gene expression and metabolic balance [146]. The mTOR pathway is a central regulator of metabolism and physiology, with important roles in the function of tissues, including liver, muscle, white and brown adipose tissue, as well as the brain [147]. Amino acids such as arginine and leucine regulate the mTOR-signalling pathway which determines neuronal complexity by regulating protein translation and actin polymerisation rates [148]. In animal studies, amino acid deficiency, in particular the absence of leucine, results in decreased synapse numbers and myelination [69]. The importance of mTOR signalling for normal brain function was emphasised by the finding that mTOR pathway dysfunction is implicated in disorders such as autism, epilepsy and neurodegenerative disorders [145]. 

Due to obligatory endogenous protein turnover, without exogenous protein, 1% to 2% of total body protein per day is lost due to proteolysis. A minimum intake of 1.5 g·kg^−1^·day^−1^ intravenous protein is required to prevent a negative nitrogen balance in preterm babies [139] and this forms the basis of current recommendations for intravenous amino acid intakes for VLBW babies (1.5 g·kg^−1^·day^−1^ increasing on day 2 to 2.5–3.5 g·kg^−1^·day^–1^ with incremental daily increases to a maximum of 3.5 g·kg^-1^·day^−1^ in the next few days) [134]. A series of studies in the 1980s and 1990s to investigate nitrogen balance in the first week after birth at differing amino acid intakes found at least 3–3.5 g·kg^−1^·day^−1^ was required to achieve nitrogen retention similar to a fetus in utero [149], with fetuses receiving at least 4 g·kg^−1^·day^−1^ amino acids via the placenta when fetal growth is most rapid [150]. Multiple studies have demonstrated that early intakes at this level result in higher protein synthesis rates, a positive nitrogen balance and are safe [151,152,153,154].

## 9. Direct Effects of Protein on Neurodevelopment 

Brain growth and development depend on a high rate of protein synthesis because protein is essentially the structural scaffolding of the brain. In animals, adequate protein supply during fetal development and the early postnatal period is essential for brain growth and myelination. The brain structures most vulnerable to inadequate protein are the cortex, hippocampus, and cerebellum [83].

Perinatal protein restriction in mice has also been found to alter the intracerebral dopamine circuit, causing altered reward-processing and hyperactivity [155]. Vucetic et al has speculated that this could be relevant to adverse neurodevelopmental effects in humans, such as attention deficit hyperactivity disorder. While this explanation of the association between neurodevelopment, behavior and protein intake in humans is speculative, protein deprivation in pregnancy and up to weaning, at least in mice, reduces brain size, dendritic arborization, cell maturation and brain connectivity of offspring [156]. However, in rats, perinatal protein restriction in the presence of neonatal hypoxia–ischaemia has been reported to have a positive impact on brain tissue loss and aversive memory impairments in inhibitory avoidance tasks [157]. In preterm babies, most, but not all, observational studies have found positive correlations between higher protein intakes, greater head circumference growth [116] and improved neurodevelopmental outcomes [109]. 

## 10. Studies of Higher vs. Lower Protein Intakes 

A Cochrane review comparing higher versus lower intakes of intravenous amino acids in newborn babies found only two randomized trials (201 participants) that could be included for the outcome of neurodevelopmental disability, although this was not the primary outcome in either trial [158]. The very low-quality evidence was insufficient to determine the effect of higher intravenous amino acid intake on neurodevelopment but did demonstrate that higher intravenous amino acid intakes reduced the incidence of postnatal growth faltering and retinopathy of prematurity without affecting mortality. One further randomized trial with the primary outcome of neurodevelopment at 2 years’ corrected age has been published since the 2018 Cochrane review and found no difference in Bayley III scores at 18–24 months’ corrected age [159]. 

A further meta-analysis that compared both low- versus high-dose and early versus late intravenous amino-acid administration in VLBW babies included 14 studies [77]. The authors concluded that the initiation of amino acids within the first 24 h after birth appeared to be safe and well tolerated, but there was no evidence of an effect on growth, morbidity or neurodevelopment. 

Thus, to date, findings from randomized trials of increased protein intake to improve neurodevelopmental outcome have been disappointing. However, considerable variability in study design, interventions, and outcomes makes meta-analysis of the available trials difficult to interpret. Furthermore, the number of participants in each trial is small, most are from single centres, and actual protein intakes were frequently lower than target, so that the trials did not achieve the intended separation in protein intakes between randomized groups (Table 1). 

One trial reported lower 18-month Bayley II indices and growth z-scores in the higher amino acid group [160,166], although by 2 years of age there was no longer any difference between the intervention and control groups. 

The lack of an effect in these randomized trials is in contrast to a cohort study in ELBW babies which reported that every 10 kcal·kg^–1^·day^–1^ increase in energy intake during the first week after birth was associated with a 4.6-point increase in MDI and each additional 1 g·kg^–1^·day^–1^ protein intake with an 8.2-point increase in MDI [117]. Since then, two further cohort studies with approximately 0.5 g·kg^-1^·day^−1^ difference in protein intake between cohorts found no effect on neurodevelopment at two years’ corrected age [167,168] and one study of 61 children (gestation <32 weeks and birthweight <1250 g) randomly assigned to receive 3.5 versus 4.8 kcal·kg^-1^·day^−1^ enteral protein reported improved neurodevelopment in the higher protein cohort at 24 months’ corrected age as assessed by Griffith Development Mental Scales [169]. This is one of the few studies in which the difference in protein intake between the higher and lower protein cohorts was > 1 g·kg^–1^·day^–1^ (1.3 g·kg^–1^·day^–1^). This study also reported significantly better growth in the subgroup of ELBW babies in the higher protein group for both length (0.16 cm per week) and head circumference (0.1 cm per week). 

Observational studies on protein intake and neurodevelopment in preterm babies are difficult to interpret because they involve various combinations of intravenous amino acids and lipids, enteral protein, increases in enteral feeds and breastmilk fortification. In addition to relatively small sample sizes, the inherent limitations of non-randomized studies allow only limited conclusions to be drawn. Nevertheless, these observational studies consistently report “real world” findings that linear growth and perhaps neurodevelopment [170] are improved or unchanged by better nutrition [167,168,169]. It is also possible that higher enteral protein rather than intravenous amino acid intakes are solely responsible for the positive findings. One of the few observational studies to include long-term follow up investigated the neurodevelopmental outcomes at 7 years of two non-contemporaneous cohorts of babies born <1500 g or <30 weeks’ gestation before and after a change in nutrition protocol to increase early intravenous and enteral protein intakes [171]. In the first week after birth, babies in the higher protein cohort received a mean of 0.4 g·kg^−1^·day^−1^ more protein, 1.8 g·kg^−1^·day^−1^ less carbohydrate and 4 kcal·kg^−1^·day^−1^ less energy than the lower protein cohort. Neurodevelopmental impairment (defined as any of Wechsler intelligence Scale for Children full scale IQ <−1 SD, Movement ABC-2 score ≤5th centile, cerebral palsy, blind or deaf) was reported in 43% of 128 children assessed, and was not different between the two cohorts. There was a non-significant trend towards more cerebral palsy in the higher protein cohort (aOR (95%CI) 7.36 (0.88–61.40), *p* = 0.07). Growth, blood pressure, glucose metabolism and body composition measures were unchanged. There were no associations between intake of any macronutrient and later cognitive impairment, but a strong positive association between early protein intake and energy to protein ratio and motor impairment and cerebral palsy. Higher intravenous amino acid intake in the first 14 days was also significantly associated with an increased risk of Movement ABC scores ≤5th centile, whereas increased enteral intakes were associated with a reduced likelihood of neurodevelopmental impairment, including motor impairment, but not cognitive impairment or cerebral palsy [172]. Interestingly, a lower energy to protein ratio in the first 14 days was associated with triple the odds of Movement ABC scores ≤5th centile and increased the odds of cerebral palsy by seven-fold, suggesting that an adequate intake of both protein and energy may be essential for optimal brain development. 

Overall, the relationship between early protein intake and later neurodevelopmental outcomes, or proxy outcomes, such as head growth or MRI findings, remains unclear and whether increased early protein intake improves or worsens later outcomes is unknown. Inclusion criteria vary widely, amino acid interventions were initiated at different postnatal ages and continued for different durations, there is variation in the protein to energy ratio and other nutrients given, e.g., lipids and electrolytes, and the difference in protein intakes between the two groups often was much less than planned. This is because achieving prescribed nutritional intakes in ELBW babies is challenging [173] and many clinicians are concerned about the potential effect of higher protein intakes on urea concentrations. For example, in the trial by Bellagama et al., amino acid intake was reduced in response to elevated urea concentrations in 23 (28%) of the high amino acid and 14 (17%) of the low amino acid group participants, (*p* = 0.09). However, the urea threshold used (>100 mg·dL^−1^) was arbitrary and not evidence-based. Associations between urea concentrations and important long-term outcomes in extremely preterm babies have yet to be established [154]. Increased plasma urea concentrations may indicate dehydration and immature kidney function or simply reflect a normal response of increased amino acid oxidation as protein accretion is achieved and a normally functioning liver and urea cycle activity detoxify ammonia [174]. 

Nevertheless, one fairly consistent finding is that a difference in protein intakes between groups of < 1 g·kg^−1^·day^−1^ is insufficient to significantly influence growth or neurodevelopment. Further large, high quality randomized trials with neurodevelopment as the primary outcome are required to answer the question of whether increased early protein intake can improve later outcomes. Pending such trials, the European Society of Paediatric Gastroenterology, Hepatology and Nutrition 2018 guidelines for VLBW babies now recommends that >3.5 g·kg^−1^·day^−1^ amino acids should only be administered as part of a clinical trial, although this was a conditional recommendation without strong consensus and the recommendations for ELBW babies from this group remain unclear [134]. 

## 11. Possible Reasons Why Research Has Not Yet Shown Whether We Can Improve Neurodevelopment with Nutrition Interventions

Several factors have an important influence on how additional protein affects neurodevelopment and have likely clouded research findings to date. 

**(i) Enteral versus intravenous protein**. The balance between enteral and intravenous amino acid may impact outcomes or may confound studies, as babies with more enteral feeding are likely to be less sick. Early introduction of enteral feeding for preterm babies is critical for the intestinal synthesis of citrulline, arginine and polyamines, as well as for intestinal motility, integrity, and growth [175]. Enteral nutritional components with immunomodulatory and/or anti-inflammatory effects may also serve as neuroprotective agents. If probiotics, prebiotic, oligosaccharides and certain amino acids can influence gut microbiota, they may also have beneficial effects on the developing brain through the microbiome-gut-brain axis. In newborn preterm pigs, the effect on neurodevelopment of 10 days of intravenous nutrition was compared with 10 days enteral nutrition [176]. Despite similar gains in body weight, the pigs who had intravenous nutrition had smaller brains, including the cerebellum, and reduced motor activity, corresponding to underdeveloped myelination, as measured by DTI. Similarly, in a study of babies born <1500 g or <31 weeks’ gestation and followed up at 7 years, higher enteral protein, fat and carbohydrate intakes in the first 7 days, and volume of breastmilk intake in the first 14 days were associated with reduced odds of neurodevelopmental impairment [171,172]. 

**(ii) The impact of energy to protein ratio**. The optimum energy to protein ratio for preterm babies is not yet known and is difficult to disentangle from the requirements for protein, carbohydrate, lipid and energy individually. Energy intake affects how protein is utilized, because when energy is inadequate, protein is oxidized to provide energy rather than being available for lean mass accretion and brain growth. Protein catabolism appears to be prevented when the energy to protein ratio is at least 34 kcal per g protein [177]. The reported nutritional intakes in the randomized trials indicate that this ratio was often not achieved (Table 1). 

**(iii) Biochemical disturbances**. High intravenous amino acid intakes in combination with low electrolytes intakes in the first few days after preterm birth may cause the characteristic biochemical disturbances known as refeeding syndrome [178]. Babies who are intrauterine growth restricted, small for gestational age, ELBW, and boys appear most vulnerable [179,180,181,182]. The proposed mechanism is that early high amino acid intakes increase insulin production and the transfer of phosphate into cells for energy and protein production, leading to hypophosphataemia, hypokalaemia, hypercalcaemia, hyperglycaemia, hypomagnesaemia and phagocyte dysfunction [183]. Refeeding syndrome is also associated with decreased serum thiamine concentrations, metabolic acidosis, hypernatraemia, hypovolaemia, ischaemia, respiratory alkalosis and chronic lung disease [178], increased sepsis [183] and higher mortality [184]. These biochemical disturbances and increased morbidities all may contribute to impaired neurodevelopment, but have not been adequately investigated to date.

**(iv) Sex differences in response to nutrients**. Girls born preterm consistently have better outcomes than boys, including neonatal morbidity [185], mortality [186] and long-term neurodevelopment [187]. Sex-related effects of preterm birth have also been demonstrated on cerebral grey and white matter volumes at 8 years of age [188]. This may be because current protein intakes meet the growth and development needs of girls but not boys, whose requirements may be higher as they are at other times of rapid growth such as adolescence. Requirements for lipids or other nutritional components may also contribute to these effects. In a recent cohort study, higher early lipid intake and enteral feed volumes were associated with improved outcome to 2 years in girls, but not boys [189]. Preterm girls were more likely to achieve survival free from neurodevelopmental impairment (147 (82%) vs. 156 (72%), *p* = 0.02). In girls, intake of lipid but not protein or carbohydrate in the first week was associated with greater odds of survival without neurodevelopmental impairment (OR (95%CI) 5.36 (2.59–12.60), *p* < 0.001). In boys, there were no significant relationships between early macronutrient intakes and survival free from neurodevelopmental impairment. This may suggest that boys need higher macronutrient intakes for optimal brain structure, or that factors other than nutrients are limiting neurodevelopment. However, few randomized trials have examined the sex-specific impact of nutritional interventions or have been powered to do so. In the future, studies to determine the effects of nutritional intakes on neurodevelopment need to be carried out to assess the effects in boys and girls separately.

## 12. Conclusions 

Optimal early nutrition providing adequate amounts of all macronutrients and micronutrients is essential for normal brain development, and enhanced nutrition in the first weeks after birth has the potential to improve neurodevelopmental outcomes. To date, there is insufficient evidence to determine the effect of higher early intravenous amino acid intake on neurodevelopment. However, few nutrition trials in extremely preterm babies have been designed to test the effect on neurodevelopment, and those that have been performed are heterogeneous in design and reporting of results. Numerous other questions relating to nutrition and its impact on neurodevelopment remain to be answered, including the optimal intravenous nutrition solutions and electrolyte intakes, the optimal energy: protein ratio, and whether girls and boys require different nutritional intakes. Techniques such as advanced MRI provide more timely, objective and specific assessments than traditional developmental screening tools and may provide useful early markers to assess nutritional interventions whilst waiting for longer-term follow-up to be completed. Future research will be of greatest value if trial designs result in separation of nutritional intakes between groups, ensure adequate energy, macronutrients and micronutrients, and are powered to assess a primary outcome of neurodevelopment separately in girls and boys. The development of minimal reporting sets and core outcome sets for nutrition research will aid future meta-analysis.

## Figures and Tables

**Figure 1 nutrients-11-02029-f001:**
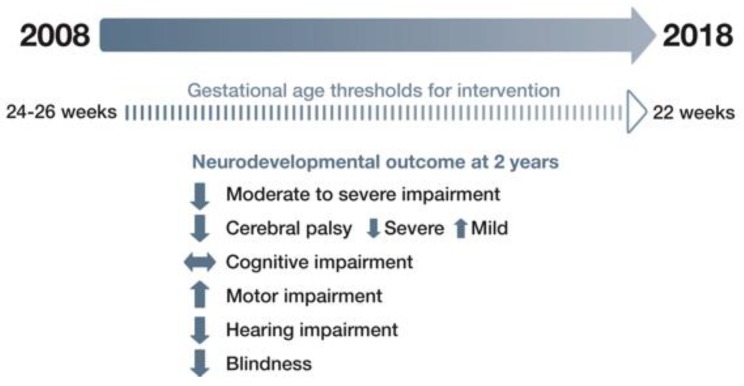
Changes in neurodevelopmental outcomes for extremely preterm babies in 2008–2018.

**Figure 2 nutrients-11-02029-f002:**
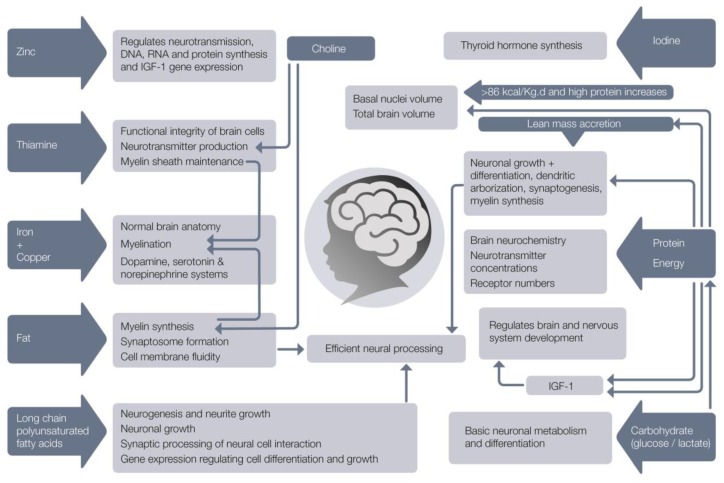
Nutrients needed for normal brain development.

**Figure 3 nutrients-11-02029-f003:**
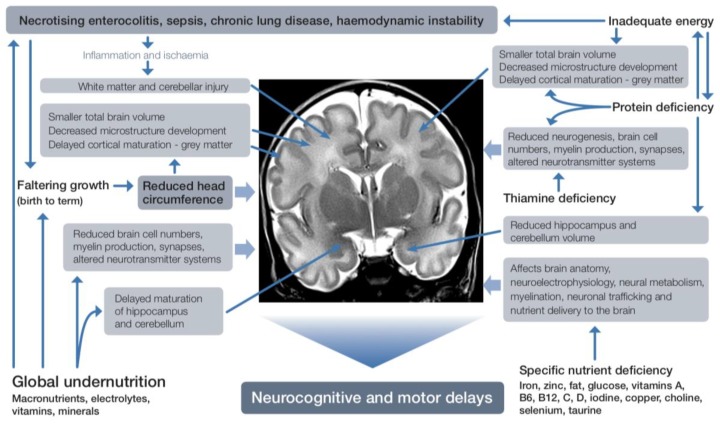
Effects of nutrient deficiency during brain development.

**Table 1 nutrients-11-02029-t001:** Randomised trials of interventions aiming for approximately 3 vs. 4 g·kg^−1^·d^−1^ IVN protein in the first week after birth.

First Author Participants Country	*n*	Protein Intake Control Group g·kg^−1^·day^−1^ *	Protein Intake Intervention Group g·kg^−1^·day^−1^ *	Amino Acid Solution	Primary Outcome	Outcomes, Conclusions and Other Details
Tan <29 weeks UK [118,137]	114	~2.3	2.6	Not specified	Head growth at 36 weeks	No difference in total brain volume on MRI at 40 weeks. Follow up rate 64% at 9 months. MDI and PDI correlated significantly with energy deficit at 28 days of age. Bayley II assessor not blinded to group allocation.
Blanco <1000 g USA [160]	32	2.9	3.7 (day 7)	Aminosyn	Mean serum potassium during first 3 days	Follow up rate 63% at 2 years. Bayley II PDI scores similar, but lower MDI score at 18 months in the high AA group. No difference at 24 months. No differences in cerebral palsy, blindness or early childhood intervention between groups at any age. More babies ≤24 weeks’ gestation allocated to higher protein group.
Burattini <1250 g Italy [161]	114	~2.5	~3.0	TrophAmine	Weight and weight <10th centile at 36 weeks	Follow up rate 87% at 2 years. No difference in Bayley III scores. Caregivers aware of group allocation.
Morgan (SCAMP) <1200 g or <29 weeksUK [116,162]	150	2.4 week 1 3.0 week 2	2.8 3.6	Vaminolact	Change in HC at 28 days postnatal age and 36 weeks	Follow up rate 62%. No difference in Bayley III assessments at 2 to 3.5 years. High protein group had higher HC z-score at 36 weeks (difference 0.40 95% CI 0.005–0.79, *p* = 0.047). Parents and assessors blinded to group allocation.
Vlaardingerbroek <1500 g Netherlands [154,163]	103	2.4 + 1.4 g lipid on day 2 or 2.4 + 2–3 g lipid from birth	3.6 + 2–3 g lipid from birth	Primène	Composite outcome of death or major disability at 2 years	Follow-up rate 92%. No difference in Bayley III assessments and incidence of major disability between groups at 2 years. At 6 weeks, weight and HC z-scores significantly higher in high protein group. Addition of lipids and extra protein increased the amount of phenylalanine used for protein synthesis.
Uthaya (NEON) <31 weeks UK [164]	133	1.7 to 2.7	3.6 from day 1	Vaminolact	Lean body mass	Follow up rate 79%. No significant differences for brain volume. HC at term-equivalent age was smaller in the high protein group (mean difference –0.8 cm, 95% CI–1.5 to–0.1 cm; *p* = 0.02), but also smaller at baseline.
Bellagamba 500–1249 g Italy [165]	160	~2.5	~3.5	TrophAmine	Weight gain from birth to 1800 g	Follow-up rate >90%. No differences in Bayley III scores at 2 years. Neonatologists not blinded to group allocation. IVN + enteral intervention. Difference in protein intakes between groups reached 1 g·kg^−1^·day^−1^ on day 4
Balakrishnan <1250 g USA [159]	168	day 1–2 day 2–2.5 day 3–3	3 3.8 4	Premasol	Bayley III at 18–24 months CGA	Follow-up rate 78%. No differences in Bayley III scores at 18–24 months. Largest study to date, single centre. Achieved at least 1 g·kg^−1^·day^−1^ difference in protein intakes between groups but for only first 3 days

Trials are listed in chronological order of publication. Ages are post-menstrual age until term-equivalent and then corrected age unless otherwise specified. * Mean actual total protein intake, AA amino acid, CGA corrected gestational age, CI confidence interval, HC head circumference, MDI Motor Development Index, MRI magnetic resonance imaging, PDI Psychomotor Development Index.

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
