# Peer review of "The Influence of Early Nutrition on Brain Growth and Neurodevelopment in Extremely Preterm Babies: A Narrative Review"

_nutrients, 2019, doi:10.3390/nu11092029_

Round 1

Reviewer 1 Report

In the present review, the Authors discuss pros and cons how nutrition, and in particular protein intake affects neurodevelopment in extremely low birth weight and extremely preterm babies.

Failure to provide the essential nutrients amounts to low birth weight and preterm babies leads to growth failure, but also increases morbidity and impairs optimal neurodevelopment. In contrast, enhanced nutrition of very preterm infants, both intravenous and enteral, promotes positive energy and protein balance and improves longer term neurodevelopmental outcomes.

Overall, this article is very well written and easy to read. The different ideas and paragraphs are well linked. The arguments of the authors are very well documented (many references). This review is very pleasant to read.

Minor point:

In the beginning of the article, the Authors could discuss in some detail the role of ketone bodies due to the very extensive experimental literature on the role of lactate, for example, as a substrate in the immature brain. They could add them to the figure 2 (Nutrients needed for normal brain development.), for example.

Line 246: extracellular could be added: “diffusion of EXTRACELLULAR water molecules”

Line 387: “Perinatal protein restriction in mice has also been found to alter the intracerebral dopamine circuit causing altered reward-processing and hyperactivity”. The effect of protein restriction would not be so clear (Effects of pre- and postnatal protein malnutrition in hypoxic-ischemic rats. Sanches EF et al., Brain Res 2012).

Reviewer 2 Report

This review is much more comprehensive than solely focusing on the influence of protein intake on neurodevelopment, as suggested in the title. In fact, the review addresses the influence of early nutrition on neurodevelopment and brain growth in extremely preterm infants, relating directly macronutrient and micronutrient intakes, and indirectly energy and macronutrient intakes, using growth and body composition as surrogates.

In any review, including narrative reviews (Baethge 2019), the type of review and the aim should be specified. Thus, I suggest a more informative title accompanied by a subtitle, for instance “The influence of early nutrition on brain growth and neurodevelopment in extremely preterm babies: a critical overview”, or the alternative subtitle “a narrative review”.

In addition, the aim of the review is lacking. In the last paragraph of Introduction, the aim and  type of review should be stated. In the Abstract (line 17) it should be stated “In this narrative review,…”.

In section 10, the need of high quality randomized trials with neurodevelopment should be stressed not only to clarify whether increased early protein intake can improve later outcomes (line 476-478), but also if it can worsen, since high protein intakes have been associated with small head circumference and poor neurodevelopment (ref. 155 and 159 of the manuscript). Therefore, I suggest stating in line 478 “…whether increased early protein intake can improve or worsen later outcomes”.
